# LC-MS/MS Based Volatile Organic Compound Biomarkers Analysis for Early Detection of Lung Cancer

**DOI:** 10.3390/cancers15041186

**Published:** 2023-02-13

**Authors:** Shuaibu Nazifi Sani, Wei Zhou, Balarabe B. Ismail, Yongkui Zhang, Zhijun Chen, Binjie Zhang, Changqian Bao, Houde Zhang, Xiaozhi Wang

**Affiliations:** 1College of Information Science & Electronic Engineering, Zhejiang University, Hangzhou 310027, China; 2Biochemical Analysis Laboratory, Breath (Hangzhou) Technology Co., Ltd., Hangzhou 310000, China; 3College of Biosystems Engineering and Food Science, Zhejiang University, Hangzhou 310058, China; 4Zhejiang Zhoushan Hospital, Zhoushan 316021, China; 5Department of Hematology, The Second Affiliated Hospital, College of Medicine Zhejiang University, Hangzhou 310009, China; 6Department Gastroenterology, Nanshan Hospital, Guandong Medical University, Shenzhen 518052, China

**Keywords:** volatile organic compounds, biomarker analysis, early screening, lung cancer, LC-MS/MS analysis, exhaled breath

## Abstract

**Simple Summary:**

In our work, we described and proposed a novel LC-MS/MS based approach for early lung cancer screening based on the VOC marker. Through this approach, two key volatile organic compounds (VOCs) used in the early screening and diagnosis of lung cancer were determined as 3-hydroxy-2-butanone and 2-pentanone. 3-hydroxy-2-butanone in the human body was found for the first time to be derived from the sugar metabolism of oral bacteria and we found a correlation between the bacteria that metabolize 3-hydroxy-2-butanone and lung cancer.

**Abstract:**

(1) Background: lung cancer is the world’s deadliest cancer, but early diagnosis helps to improve the cure rate and thus reduce the mortality rate. Annual low-dose computed tomography (LD-CT) screening is an efficient lung cancer-screening program for a high-risk population. However, LD-CT has often been characterized by a higher degree of false-positive results. To meet these challenges, a volatolomic approach, in particular, the breath volatile organic compounds (VOCs) fingerprint analysis, has recently received increased attention for its application in early lung cancer screening thanks to its convenience, non-invasiveness, and being well tolerated by patients. (2) Methods: a LC-MS/MS-based volatolomics analysis was carried out according to P/N 5046800 standard based breath analysis of VOC as novel cancer biomarkers for distinguishing early-stage lung cancer from the healthy control group. The discriminatory accuracy of identified VOCs was assessed using subject work characterization and a random forest risk prediction model. (3) Results: the proposed technique has good performance compared with existing approaches, the differences between the exhaled VOCs of the early lung cancer patients before operation, three to seven days after the operation, as well as four to six weeks after operation under fasting and 1 h after the meal were compared with the healthy controls. The results showed that only 1 h after a meal, the concentration of seven VOCs, including 3-hydroxy-2-butanone (TG-4), glycolaldehyde (TG-7), 2-pentanone (TG-8), acrolein (TG-11), nonaldehyde (TG-19), decanal (TG-20), and crotonaldehyde (TG-22), differ significantly between lung cancer patients and control, with the invasive adenocarcinoma of the lung (IAC) having the most significant difference. (4) Conclusions: this novel, non-invasive approach can improve the detection rate of early lung cancer, and LC-MS/MS-based breath analysis could be a promising method for clinical application.

## 1. Introduction

Lung cancer is among the top three cancers affecting individuals irrespective of gender and is responsible for cancer-related deaths globally. However, more than two-thirds of lung cancer incidences are often discovered at late and incurable stages due to early-stage unrecognizable symptoms. Nevertheless, only 4.7% of patients with metastatic stage IV lung cancer survive for five years, compared to 56.3% of patients with stage I cancer [1,2]. Early diagnosis and prompt surgery help to improve the cure rate and thus reduce the mortality rate. Annual low-dose computed tomography (LD-CT) screening is an efficient lung cancer-screening program for a high-risk population [3,4]. However, LD-CT has often been characterized by a higher degree of false-positive results and an elevated risk of misdiagnosis. To meet these challenges, volatolomics, which encompass the analysis of volatile organic compounds (VOCs) emanating from cancer cells that can be detected in various body fluids, including blood, urine, exhaled breath, and sweat, presents an auspicious development in cancer biomarker-based approaches. The main characteristic of cancerous tumor is the uncontrolled growing of cells inside the organism, changes in the body’s metabolism, and the outrush of VOC emissions. Likely, these uncontrolled growths are, consequently, changes in the metabolism that could be analytically monitored [5]. Depending on where the cancer cells are located, they provide different characteristic profiles. These profiles, as fingerprints, are used for differentiation in a comparison to normal cells [6]. In particular, the breath VOC fingerprint analysis has recently received increased attention for its application in early lung cancer screening thanks to its convenience, non-invasiveness, and being well tolerated by patients [7]. Inspired by the unusual smell of diabetes, liver cirrhosis, renal failure, and other diseases, S. M. Gordan successfully detected VOCs in exhaled breath to diagnose lung cancer in 1985. The results showed that the VOCs in the exhaled breath of 12 patients with advanced lung cancer were different from those of 17 healthy people, and the diagnostic model constructed by acetone, methyl ethyl ketone, and *n*-propanol can completely distinguish the lung cancer group from the normal group [8]. This pioneering research using VOC fingerprint analysis to diagnose lung cancer has gradually attracted more researchers’ attention. Besides, it has been explored in diagnosing other cancers and infectious diseases, such as breast cancer, COPD, asthma, etc. [9,10,11,12,13]. 

So far, about 83-lung cancer VOCs have been identified [14,15]. However, no single VOC or VOC fingerprint model has been approved for clinical application yet due to several shortcomings, including poor reproducibility, making large-scale verification very difficult. However, previous results suggested that, for the same illness, the repetitiveness of VOC markers screened between different laboratories or even in the same laboratory at different periods was relatively low [16,17]. This resulted in the absence of a unified standard for sample processing, instrumentation, data analysis, and quantitative verification method [18]. Mass spectrometry (MS) has emerged as the most common detection technique widely used to detect VOCs. The LC-MS/MS system, in particular, is a powerful analytical tool for the detection, confirmatory identification, and quantitation of various molecules with myriad applications in numerous research fields [19]. The LC-MS/MS system comprised three key parts: an ion source, mass analyzer, and a detector with electrospray ionization (ESI), atmospheric pressure ionization, and atmospheric pressure chemical ionization (APCI) being the commonly used ion sources. Compared with GC-MS, the LC-MS-based analysis is affordable, and it requires less complicated sample preparations, and it possesses high resolution and fast analyte detection speed. Besides, the number of metabolites identified by the GC-MS study are limited to about 100, while LC-MS/MS could identify nearly 500 metabolites. Besides, LC-MS/MS techniques has demonstrated high separation efficiency for thousands of metabolites in various samples without loss of sensitivity [20,21]. Despite the wide application of mass spectrometry, for detecting VOCs, its application in breath VOCs fingerprint analysis for early detection of lung cancer biomarkers has not been reported thus far. 

In the present study, we hypothesized that clarifying the source of VOCs and confirming their correlation with tumors is a prerequisite for establishing repeatable and standardized testing and diagnosis. A standard approach could be used to screen early lung cancer patients based on VOC biomarkers using a novel sensing technique, liquid chromatography with tandem mass spectrometry (LC-MS/MS), to enhance the sensitivity and reliability of qualitative results. The detection objects are 25 carbonyl VOCs with stable derivatization reactions and LC-MS/MS analysis methods, and 4 carbonyl VOCs cannot be detected and were not displayed (Appendix A). In addition to looking for differences in the exhaled breath of lung cancer patients, we also studied the possible metabolic sources and metabolic mechanisms of 3-Hydroxy-2-butanone and 2-Pentanone (TG-4 and TG-8) in the exhaled breath of lung cancer patients, and we found that TG-4 is mainly derived from the sugar metabolism of oral bacteria in lung cancer patients, and TG-8 may be derived from lipid metabolism in the human body. The changes in oral bacteria in lung cancer patients were analyzed by 16SrRNA”.

## 2. Materials and Methods

### 2.1. Materials 

2,4-Dinitrophenylhydrazine (DNPH) was purchased from Rhawn (Shanghai, China), with purity ≥ 99%. Methanol and acetonitrile of liquid chromatographic grade (purity ≥ 98%) were purchased from Sigma-Aldrich (Shanghai, China). Hydrochloric acid and silica gel powder of 300–400 mesh were acquired from Sinopharm Chemical Reagent (Beijing, China). The standard solution was composed of: 2-butanone-DNPH, butyraldehyde-DNPH, pentanal-DNPH, pentanone-DNPH, acrolein-DNPH, acetaldehyde-DNPH, butenal-DNPH, furfural-DNPH, formaldehyde-DNPH, hexaldehyde-DNPH, benzaldehyde-DNPH, 2-crotonaldehyde-DNPH, formylofuran-DNPH, heptanal-DNPH, octanal-DNPH, nonanal-DNPH, decanal-DNPH, and cyclohexanone-DNPH, which were purchased from A Chemtek; additionally, acetoin-DNPH, 2-pentanone-DNPH, and 4-heptanone-DNPH, with a purity of above 98%, were acquired from TanMo Quality Testing Co., Ltd. (Beijing, China). A blank solid-phase extraction (SPE) column without any other filler applied was purchased from Biocomma Limited Co., Ltd. (Shenzhen, China). A 5 L Tedlar bag was purchased from Sigma-Aldrich (Shanghai, China), and prior to their use, bags were cleaned with N_2_ at a high temperature (80 °C) until carbonyl compounds were free. Ultrapure water was used for all aqueous solution preparation. A detailed procedure for the solid phase column extraction is presented in Appendix A.

### 2.2. Study Participants

Breath samples were collected from the cardiothoracic surgery department of Zhoushan hospital (Zhoushan city, Zhejiang Province, China). Approval was obtained from the ethics committee of the Zhoushan hospital (2019/208). All individuals involved in the study were formally enrolled, signed written documents, and interviewed regarding health records and smoking habits. All lung cancer patients were in the early stage of lung cancer and were diagnosed by imaging techniques (computed tomography and positron emission tomography) and histology (biopsy). Participants with a history of malignancy and ventilation dysfunction were excluded. Breath samples (fasting and 1 h after a meal) were collected from lung cancer patients before surgery, three to seven days after surgery, and four to six weeks after surgery. The control group consisted of healthy volunteers without pulmonary disease history. Subjects were asked not to engage in vigorous physical activity before sample collection for two days. After dinner on the night before collection, subjects were asked to abstain from food and drink (except water) and smoking. During sampling, the volunteers blew air into the Tedlar bags (>3 L) while sitting in a specific chamber. 

### 2.3. Breath Sample Processing and Analysis 

During the sampling process, the volunteer sits in the sampling room and blows into the sampling bag (>3 L). The bags were stored at ambient temperature (22 ± 4 °C) in the dark, and the analysis was conducted within 24 h after collection, minimizing contamination or loss of compounds. During treatment, the sampling bag and DNPH-coated silica gel solid-phase extraction column were placed in an oven at 50 °C for 30 min. Then, the air outlet of the sampling bag was connected with the lower end of the adsorption tube using a catheter, and the upper end of the DNPH-coated silica gel solid-phase extraction column was connected to the suction device. The pumping rate was set at 50 mL/min, and the pumping volume was set at 3 L. After the extraction, 2 mL acetonitrile was added from the adsorption tube’s upper end in successive drops. After natural or pressurized elution, the eluent was collected, diluted to 2 mL with acetonitrile, and filtered using a 0.22 μm PTFE needle filter to prepare the sample.

### 2.4. Liquid Chromatography-Tandem Mass Spectrometry (LC-MS/MS) Analysis

The LC-MS/MS analysis was carried out according to P/N 5046800 standard using (AB SCIEX 4500MD Mass Spectrometer, Framingham, MA, USA). The separation and analysis conditions were as follows: Thermo column-AcclaimTM Carbonyl C18 (3 μm, 120 Å, 3.0 × 150 mm); mobile phase: 5 mmol tendency for ammonium acetic acid aqueous solution (A)—acetonitrile (B); gradient elution program: 0–3 min, 50% B, 3–5 min, 50–60% B, 5–8 min, 60–70% B, 8–9.5 min, 70% B, 9.5–10 min, 70–80% B, 10–11 min, 80% B, 11–11.5 min, 80–85% B, 11.5–12.5 min, 85% B, 12.5–13 min, 85–90% B, 13–14.5 min, 90% B, 14.5–15.5 min, 90–95% B, 15.5–16 min, 95% B, 16–18 min, 95–50% B, 18–20 min. 50% B. Total flow rate: 0.4 mL/min; column temperature: constant temperature 40 °C; sample room temperature: constant temperature 10 °C.

A negative mode was used for operating the electrospray source. The curtain, nebulizer, auxiliary, and collision gas (nitrogen) were used. The ion transfer voltage and the probe temperature parameter were set to −4500 V and 500 °C, respectively. Sample analysis was carried out in MRM mode, with a dwell time of 25 ms per channel. The infusion of standard solutions and source-dependent parameter optimizations were performed. The most and second-most sensitive fragment signals were selected as the quantification and confirmation ion pairs. The detailed parameters of the electrospray source for optimization analysis are summarized in Appendix A.

### 2.5. Quality Control

Experimental materials, including columns, were sourced in bulk from a single lot. Reagents and consumables were tested prior to use. The quality control samples were spaced equally among the injections for each day, and all experimental samples were arbitrarily distributed throughout each day’s run. To monitor data quality and process variation, a plurality of quality control points was arranged in parallel and evenly distributed among the experimental samples. Through injection of an authentic standard, limit of detection (LOD), limit of quantification (LOQ), and Relative Standard Deviation (RSD) analyses, the proposed High-performance liquid chromatography (HPLC) quantitative analysis method was verified. Linear regression with the peak area (Y) of each analyte and the corresponding mass concentration (X, μg/L) was performed to obtain the standard curve and correlation coefficient (R) of 21 VOCs. A good linearity of each calibration curve was achieved (R >0.99). The LOD and LOQ are shown in Appendix A based on the content of the target compound when the characteristic ion chromatographic peak S/N ≥ 3. The RSD values of intra-day and inter-day variability, repeatability, and stability of 21 VOCs were all less than 5%. The results show that the established method can be used for the determination of 21 VOCs in breath samples.

### 2.6. 16S rRNA

Saliva samples for genomic DNA extraction were randomly collected from some participants before breakfast, gargled with normal saline at 15 mL, collected into a 50 mL conical tube, and stored at −80 °C until further processing. Genomic DNA was extracted for HiSeq PE250 sequencing. Based on the Illumina HiSeq sequencing platform, according to the amplified 16S region characteristics, paired-end sequencing was used to construct a small fragment library for paired-end sequencing. Genome analysis was performed based on sequencing results.

### 2.7. Statistical Analysis

GraphPad Prism 7 and SPSS 22 were used for graphic presentation and statistical analysis. *t*-tests were used to compare the characteristic of the study population of lung cancer patients with healthy controls. The data are presented as the mean ± standard error, and the *p*-value of <0.05 was used to assign statistical significance. The differences in VOC concentrations between the different groups were shown using box and whisker plots, and the statistical significance of the differences was assessed using a non-parametric Kruskal-Wallis test to count the non-normality of the distribution. To evaluate the differentiation performance (specificity and sensitivity), the area under the curve (AUC) of the receiver operating characteristic (ROC) curves was used. A diagnostic prediction model was created for early lung cancer versus controls using a decision tree—random forest analysis based on the subset of VOCs that are statistically relevant for discrimination.

The lung cancer prediction model was established based on the random forest algorithm, and indicators, such as gender, age, and VOCs, were selected as model features. The data set is divided into a training set (1/2) and a validation set (1/2). Among them, the out-of-bag data generated during the random forest training process, which account for about 1/3 of the training set, are used as the test sets. Input the training set data into the random forest model modeling (based on the sklearn framework); the prediction accuracy of the output of random forest out-bag test set is used as input for the training set data in the random forest model modelling. Input the test set data into the built model, and output the test set model confusion matrix to predict its sensitivity and specificity.

## 3. Results

### 3.1. Effect of Ambient Air

In this study, the long-term dynamic changes in the environment and the effect of respiration on volunteers were investigated. All human exhaled air samples in this experiment are from oral exhalation. A total of 21 carbonyl VOCs were unambiguously identified and quantified by LC-MS/MS analysis. The differences between exhaled breath and ambient air were shown in Appendix A. Among the 21 carbonyl VOCs, 10 carbonyl VOCs in the environment show higher concentrations than the human exhaled breath, which may impact the discriminant analysis of cancer patients’ exhaled breath.

Moreover, the concentrations of TG-1, TG-24, and TG-25 in different locations were different during the experiment. The concentrations of TG-1 and TG-24 were also different in the same place at different times (Appendix A). Besides, TG-24 at varying concentrations was found in different brands of syringes and organic solvents used in the experiment. This was consistent with previous reports that TG-24 and TG-25 are released in some medical equipment and pipelines [22]. Therefore, TG-1, TG-24, and TG-25 were excluded to avoid interference from ambient air and medical instruments. The concentration of 11 carbonyls VOCs in the environment is similar to or lower than that of human exhaled breath. Among them, TG-4, TG-8, TG-10, and TG-13 in human exhaled breath were significantly higher than in ambient air. Therefore, the 11 carbonyl VOCs were analyzed and finally assessed for their potential relationship with lung cancer.

### 3.2. Carbonyl VOCs in Exhaled Breath of Early Lung Cancer Patients 

Through the pre-experiment involving 40 lung cancer patients and 40 control individuals, exhaled breath was collected at fasting, 1 h after the meal, and 2 h after the meal. The results showed that compared with the control group, TG-4 in the exhaled breath 1 h after the meal was significantly higher among lung cancer patients, and TG-8 was significantly higher at any time (Appendix A). Therefore, subsequent experiments sampled exhaled breath for VOC analysis after morning fasting and 1 h after the meal from early lung cancer patients before surgery, three to seven days after surgery, and four to six weeks after surgery. A total of 386 people participated in the 17-month study: 291 early lung cancer patients and 95 healthy controls. Table 1 summarizes the clinical characteristics of this population. The lung cancer group and the control group differ significantly (*p* < 0.05) according to age and gender. However, the smoking habit was not significantly different among the sample population.

The difference in carbonyl VOCs in the exhaled breath between the control group and early lung cancer patients before surgery, three to seven days after surgery, and four to six weeks after surgery was mainly at 1 h after the meal. The statistical data are summarized in Appendix A and the results in Figure 1 showed significant differences in the concentrations of 7 VOCs. TG-4 and TG-8 in the exhaled breath of patients with early lung cancer were significantly higher before surgery but decreased after surgery. Nevertheless, both before and after surgery, TG-7, TG-11, TG-19, TG-20, and TG-22 were continuously low. Figure 1 depicts the ROC curve analysis results of the 7 VOCs. It is clear that TG-4 was the VOC with the most significant difference, and its sensitivity and specificity were 70% and 76%, respectively. However, the sensitivity and specificity of a single marker for screening are not yet sufficient for clinical applications. Therefore, through the comprehensive analysis of seven markers (decision tree-random forest discriminant analysis), the sensitivity and specificity were 83% and 78%, respectively. By ignoring the environmental impact, the sensitivity and specificity for the comprehensive analysis of 21 detectable VOCs were 97% and 100%, respectively.

Early lung cancer manifestations are divided into different stages: adenocarcinoma in situ (AIS), minimally invasive adenocarcinoma (MIA), or invasive adenocarcinoma (IAC). The AUC for the ROC curves to discriminate AIS/MIA/IAC from controls were shown in Appendix A and Figure 2. Approximately 86% of patients with early lung cancer have lung adenocarcinoma, and early lung adenocarcinoma was also divided into different stages of AIS/MIA/IAC. The AUC value of ROC curve analysis to distinguish AIS/MIA/IAC from the control group is shown in Appendix A. Compared with the control group, the difference in IAC was the most significant. The sensitivity and specificity of the random forest discriminant analysis of seven markers were 96% and 73%, respectively. Ignoring the environmental impact, the sensitivity and specificity of the random forest discriminant analysis of 21 detectable VOCs were 96% and 86%, respectively. 

### 3.3. Effects of Diet on Exhaled Breath Carbonyl VOCs

Diet specifically affects carbonyl VOC concentration in human exhaled breath, but studies on how it affects the carbonyl VOCs in human exhaled breath are scarce. Therefore, in this study, changes in carbonyl VOC concentration in human exhaled breath after eating some common foods (the methods and a list of foods are shown in Appendix A) were investigated. The results showed that eating protein and lipids does not affect the concentration of 11 carbonyls VOCs in the exhaled breath. However, eating sugar-containing foods causes a significant change in the concentration of TG-4 in the exhaled breath (Figure 3A–C). Besides, after eating a banana, the concentration of TG-8 in the exhaled breath also increased, with the same trend as TG-4, returning to normal levels within 1 h. After drinking 50 mL of a characteristic Chinese liquor, TG-8 and TG-10 in the exhaled breath can reach high concentrations within 5 min and fail to return to normal levels within 2 h.

Because of the change in TG-4 in exhaled breath after eating sugar-containing foods, the changes in exhaled breath after drinking a variety of sugar solutions were studied (Figure 3D). The results showed that, after drinking different sugar solutions, the TG-4 in the exhaled breath reached a high concentration at 5 min, decreased significantly at 30 min, and recovered to the fasting level within 1 h. According to Pearson correlation analysis, the concentration of TG-4 from the exhaled breath after 5 min of drinking sugar solution was positively correlated with the glycemic index (GI) (r = 0.975, *p* = 0.005, Appendix A). It is worth noting that short-term protein and lipid diets have no effect on exhaled breath, but after long-term protein and lipid diets (ketogenic diet), TG-8 and TG-14 concentrations in exhaled breath remained high and decreased only after carbohydrate intake (Figure 3E,F, see the Appendix A for the experimental method of dietary influence). Therefore, volunteers whose breath was collected needed to regulate their diet.

### 3.4. Influence of Exhalation Mode on Exhalation Collection

Based on the earlier results on the influence of diet on exhaled air, we speculated that the carbonyl VOCs in the exhaled breath might come from the oral cavity. Therefore, we chose the exhalation collection of the oral cavity and nasal cavity after drinking a 5% glucose solution for verification. The results (Figure 4) suggest that TG-4 and TG-10 concentrations in the oral exhaled breath were significantly higher than in the nasal cavity, regardless of status before or after drinking sugar, indicating that carbonyl VOCs in oral metabolism were the main components in exhaled breath. Besides, the significantly higher TG-4 in oral exhalation after drinking sugar or oral sugar indicated that TG-4 largely comes from the oral cavity.

### 3.5. Mouth Glucose Metabolism

The human mouth houses more than 700 bacterial species, some of which can be metabolized to produce TG-4. The variation in characteristics of TG-4 concentration in the exhaled air was observed after taking sugar in the mouth for 30 s (Appendix A). The concentration of TG-4 in the exhaled breath for all sample populations had an initial increase and then decreased after sugar intake for 30 s, reaching the highest concentration between 5 and 20 min (Appendix A). Compared with direct drinking of glucose solution, the concentration of TG-4 in exhaled breath increased significantly after sugar intake for 30 s, indicating that TG-4 came from the sugar metabolism of oral microorganisms.

### 3.6. 16S rRNA Detection

To investigate the bacteria in the oral cavity, we randomly sampled the oral saline gargle of 20 lung cancer patients and performed gene sequencing and Spearman correlation coefficient analysis. By testing the composition of microorganisms in oral saliva, the study found that, compared with the healthy control group at the phylum level, the relative abundance of *Elusimicrobia* decreased significantly (Figure 5). At the genera level, lung cancer patients have a significantly higher relative abundance of *Desulfobulbus, Family_xiII_UCG-001, Gemella, Halomonas, Klebsiella, Roseburia, Ruminiclostridi_5, Synergistes,* and *Alloscardovia* in their oral saliva than the control group. In contrast, *Alloprevotella’s* relative abundance was significantly lower (Figure 5). 

The results showed that the relative abundance of *Streptococcus pharyngitis, Porphyromonas pulposus, Lactobacillus mucosa,* and *Alloscardovia_omnicolens* positively correlated with TG-4 in the exhaled breath. Moreover, the relative abundance of TM7 animal phylum oral clone DR034 was negatively correlated with exhaled breath TG4 (Table 2), which signaled that TG-4 has a specific correlation with bacterial abundance in the oral saliva. However, how oral bacteria affect the concentration of TG-4 in exhaled breath remains to be determined.

## 4. Discussion

The use of breath analysis for cancer diagnosis has been around for more than 30 years since S. M. Gordan’s pioneering research in 1985. The lack of standard procedures for breath analysis is responsible for the results’ variability and poor repeatability. Table 3 compares the most commonly used VOC biomarker detection techniques from the exhaled breath of lung cancer patients. Gas chromatography coupled mass spectrometry (GC-MS) has been widely used in many research groups [23,24,25,26,27,28,29]. Despite being a powerful tool for VOC analyses, its application has been limited by the complex sample preparation and time-consuming analysis, the requirement of trained personnel, and high cost. The other commonly used technique, the headspace temperature programmed vaporization mass spectrometer (HS-PTV-MS), provides a lower limit of detection besides spectral overlap, making compounds’ identification difficult [30]. Electronic noses, such as gold nanoparticles (GNP), are based on chemical reactions with sensor materials. They provide highly sensitive results that are easy to interpret, but quantitative results cannot be obtained due to the inability to identify unknown substances [31]. Another promising method [32] for the detection of early lung cancer is using FT-ICR-MS, which significantly elevated the concentration of the detected VOC in lung cancer patients; a statistical classification model was used to analyze how the measured concentration of the carbonyl compound can be used to determine the compounds that are best for distinguishing between lung cancer patients from healthy controls. FT-ICR-MS were used to capture and clarify the molecular formulae of carbonyl compounds (hydroxyl-acetaldehyde and 3- hydroxy-2-butanone) of the trace VOC from exhaled breath in combination with GC-MS for the analysis of the detected VOC [33]. Liquid chromatography and tandem mass spectrometry (LC-MS/MS) are sequential analysis platforms that utilize high-performance liquid chromatography as the separating technique and high-resolution mass spectrometry as the detection system. LC-MS/MS is more suitable for analyzing metabolites with low volatility or poor thermal stability than the above and other analytical techniques.

Multiple candidate markers have been discovered, but they have not yet entered clinical applications. Their common shortcoming was low repeatability, and the underlying mechanism for the observed difference has not been analyzed and discussed in depth. Exhalation analysis is affected by many factors, such as the exhalation environment, exhalation method, daily diet, and internal factors [34]. The current study, therefore, uniquely examined their influence in different ways, analyzed the source of their differences, and devised means to avoid the influence of these differences. Ambient air pollution has a major impact on respiratory analysis. As such, a number of approaches were proposed to reduce the local background: inhaling clean air, balancing ambient air, collecting background samples, and calculating alveolar gradients or setting cut-offs [35,36,37]. We chose to collect the air in the sampling environment for a long time and found that the environmental changes in the same place are in a fluctuating stable balance. However, TG-1, TG-24, and TG-25 in ambient air varied greatly and were previously used as markers for disease diagnosis [38]. TG-1 could come from tobacco burning, building materials, and some unknown sources. TG-24 and TG-25 could originate from organic solvents, disinfectants, syringes, hoses, and other experimental and medical supplies. For example, the variation in TG-24 concentration for acetonitrile solvents of different brands can reach 1.5–3.4 fold, while the difference in acetonitrile solvents of the same brand, but being parts of different batches, could be up to 2–7.5 fold. Therefore, their high concentrations for different reasons may lead to different test results. Therefore, to ensure the reliability of the results, we have excluded VOCs in the exhaled breath that are lower than the environment, but these VOCs are not unchanged in the exhaled breath. The high sensitivity and specificity of the random forest discriminant analysis results of 21 VOCs indicated that the VOCs in the exhaled breath lower than the environment are also meaningful. In order to ensure the stability of the exhalation collection, the exhalation collection method is also very important. Some studies used nasal intubation for exhaling, and one-lung ventilation was used during lung tumor resection, but such collection methods are invasive and can cause physical discomfort [39,40]. A variety of non-invasive and convenient sampling methods were tried in this study, such as nasal/oral breath sampling, back-end air sampling, clean room sampling, breath-holding sampling, etc. Interestingly, no significant difference between the selected markers was found, which differs from the conclusions in some papers [38]. It is speculated that the larger number of exhalations collected (>3 L) in the current study was useful in keeping the total amount of markers relatively high, as well as reducing the differences between several exhalation collection methods and the different populations’ breathing patterns. Currently, there is little data on respiratory carbonyl VOCs associated with dietary regimes [41]. The results on the effects of the daily diet show that the complexity of the diet can also cause changes in some VOCs in the exhaled breath, and a long-term protein and lipid diet can cause changes in the TG-8 and TG-14 in the exhaled breath. Eating high-sugar foods can also cause changes in some VOCs in the exhaled breath, but they return to normal levels within an hour. There is a significant difference between the exhaled breath of early lung cancer patients and the control group 1 h after eating. Therefore, changes in VOCs in the exhaled breath of lung cancer patients after a unified diet are observed. Strenuous exercise also has a certain impact on the analysis of exhaled breath. After strenuous aerobic or anaerobic exercise, the metabolism of glycogen and fat in the body accelerates, and its metabolites can be excreted through urine, sweat, and exhaled breath. Therefore, strenuous exercise was prohibited two days before sampling [42].

The reliable data of the breath detection analysis were obtained through a series of experiments by eliminating the interfering factors and their influence. Compared with the control group, the seven VOCs with the most significant differences in early lung cancer patients’ exhaled breath were TG-4, TG-7, TG-8, TG-11, TG-19, TG-20, and TG-22. TG-4 and TG-8 in the exhaled breath of patients with early-stage lung cancer before surgery were significantly higher but decreased after surgery. However, regardless of before or after surgery, TG-7, TG-11, TG-19, TG-20, and TG-22 were all low. TG-7, TG-11, TG-19, TG-20, and TG-22 belong to the aldehyde group of VOCs derived from several source and pathways, including metabolized alcohols, hydroperoxide reduction by cytochrome p450 as a secondary product of lipid peroxidation, tobacco smoke (saturated and unsaturated aldehydes, e.g., formaldehyde) and detoxification processes of tobacco by-products, and dietary sources [43]. The low level of these aldehydes in lung cancer patients’ exhaled breath may be because the tumor inhibits the metabolic pathway of certain aldehyde substances, while the inhibitory effect is sustained and long, and surgical resection of the tumor cannot restore its normal metabolism. 

TG-4 and TG-8 are members of the ketone group of VOCs produced in the liver from fatty acids, such as acetone, and they are then oxidized in the Krebs cycle by peripheral tissues. The liver produces acetone, a common human VOC, by decarboxylating surplus acetoacetate from acetyl-CoA. Diet also has an impact on ketone levels in the blood, increasing with the increase in protein or fat metabolism (e.g., notably in cachexia). Based on the possible causes of ketones and the results of ketogenic experiments, we speculated that TG-8 might be mainly produced by fat metabolism in the human body, which is closely related to occurrence and metastasis of tumors. Metabolism in the tumor cell metabolism is far more vigorous than a normal cell. Cancer-related fat cells can produce fatty acids through lipolysis, produced by decomposition. They are delivered to tumor cells as energy sources, and, at the same time, adipokines are secreted to promote tumor cells’ adhesion and metastasis. Therefore, it is not surprising to observe an increased TG-8 concentration in the exhaled breath of lung cancer patients. The related metabolic mechanism can return to normal after tumor resection, and the concentration of TG-8 in the exhaled breath also begins to recover.

Compared with the control group, TG-4 was the most different VOC marker in the exhaled breath of the lung cancer group. There was no obvious change in the ketogenic diet experiment, and fat metabolism could not very well explain its primary source. According to the experimental results of oral glucose metabolism, we speculated that TG-4 came from the glucose metabolism of oral bacteria. The results of 16SrRNA analysis showed that the relative abundance of *Desulfobulbus, Family_xiII_UCG-001, Gemella, Halomonas, Klebsiella, Roseburia, Ruminiclostridi_5, Synergistes,* and *Alloscardovia* in the oral saliva of lung cancer patients were significantly higher than that of the control group. Therefore, the increased concentration of TG-4 in the exhaled breath of lung cancer patients was probably due to obvious changes in TG-4 metabolizing bacteria in the oral cavity of lung cancer patients. According to existing reports, the microorganisms producing acetoin in the body comprise mainly bacteria, such as *Enterobacter, Bacillus, Klebsiella, Lactococcus, Paenibacillus, Serratia, Acetobacter, Aeromonas,* etc. [44,45,46]. Hence, we speculate that *Klebsiella* may be the main genus that metabolizes TG-4 in the oral cavity of lung cancer patients. The TG-4 in the exhaled breath of lung cancer patients was higher than that in the control group, and it was statistically significant. It may be due to tumor resection and drug treatment to eliminate some related bacteria, so the TG-4 concentration in the exhaled breath of lung cancer patients decreased after surgery. Many studies have proven that oral bacteria in cancer patients are related to cancer occurrence and development. *Porphyromonas gingivalis*, a pathogen of periodontitis, has been proven to promote the occurrence and development of colorectal cancer. The bacteria have also been found in the oral cavity of lung cancer patients. Besides, its abundance positively correlated with TG-4 [47]. Many studies have proven that the structure of the salivary flora of lung cancer patients is similar to that of the lungs [48,49]. Due to time and resource constraints, the sample size of this analysis is small, and it was not possible to analyze whether the same changes have occurred in the lung flora of patients with lung cancer. Therefore, whether the bacteria in the mouth and lungs of lung cancer patients (especially the TG-4 metabolizing bacteria) have the same changes remains to be determined.

The limitations of the current study include fewer samples in the control group, age and gender distribution between control group and lung, and the high detection rate of early lung cancer by breath analysis, which may all be related to the small sample size of the control group. Nevertheless, what cannot be disputed is the difference in VOC trends observed in the exhaled breath between the control group and the early lung cancer group. In comparing the different stages of early lung adenocarcinoma and the control group, the difference in the IAC stage was the most obvious. Using seven significantly different VOC screening analysis methods, IAC has higher sensitivity than all early lung cancer group analyses, indicating that the concentration of VOCs in the exhaled breath is related to the development of lung lesions. Moreover, due to the degree of lung adenocarcinoma infiltration, the change in VOCs in the exhaled breath will become more apparent, and the breath analysis has a higher detection rate for early lung adenocarcinoma.

## 5. Conclusions

The current study presented a standardized approach that could be used to screen early lung cancer patients based on VOC markers, overcoming the sensitivity and reproducibility challenges associated with conventional testing methods. Using a developed LC-MS/MS acquisition method, the VOC biomarker test may, in the future, be complementary to the imaging, sputum cytology, and biopsy tests following validation using prospective studies. For each lung cancer patient, VOC analysis could be expressed as a chance of discovering the disease early. Nevertheless, although breath VOC analysis could represent a highly probable method to discover lung cancer, yet, it is not conclusive and would require an optimum combination of sensitivity and specificity. In this article, in addition to looking for differences in the exhaled breath of lung cancer patients, we also studied the possible metabolic sources and metabolic mechanisms of TG-4 and TG-8 in the exhaled breath of lung cancer patients. We found that TG-4 is mainly derived from the sugar metabolism of oral bacteria in lung cancer patients, and TG-8 may be derived from lipid metabolism in the human body. The changes in oral bacteria in lung cancer patients were analyzed by 16SrRNA. The sensitivity and specificity of TG-4, with the most significant difference in receiver operating characteristic (ROC) curve analysis, were 70% and 76%, respectively. This novel, non-invasive approach can improve the detection rate of early lung cancer, and further work to improve the suitability of this technique and to assess its limitations in clinical practice would be required.

## Figures and Tables

**Figure 1 cancers-15-01186-f001:**
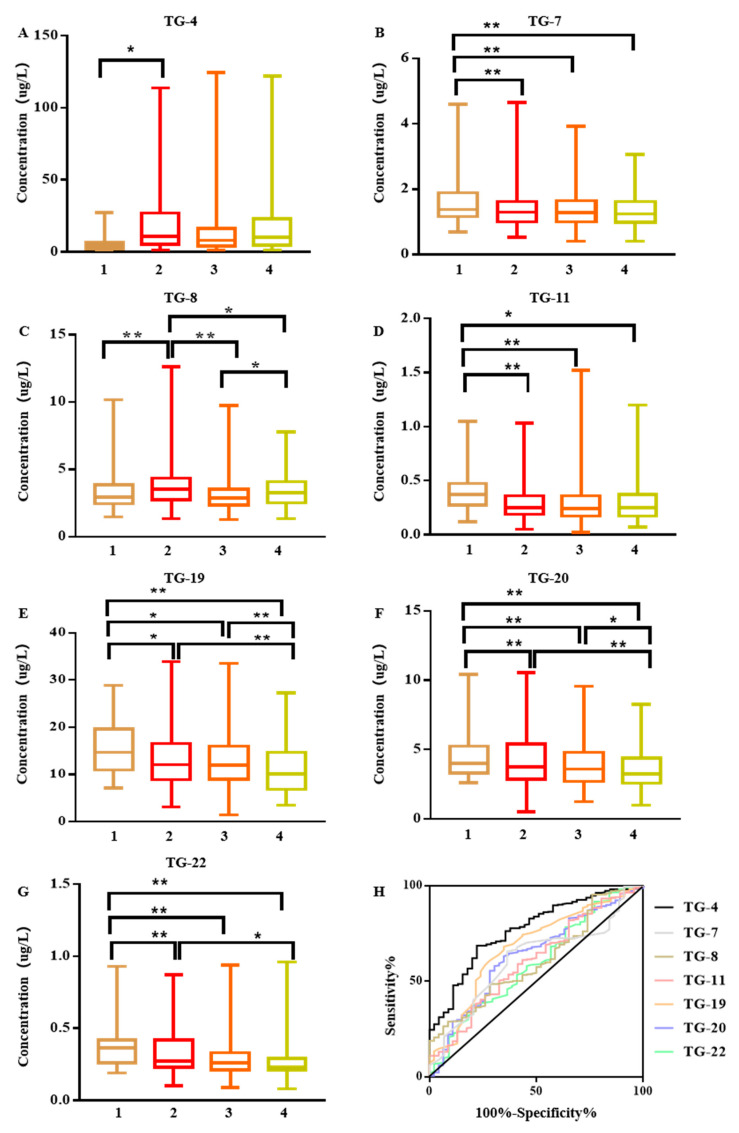
(**A**–**G**), 1, control; 2, lung cancer before the operation; 3, lung cancer after the operation after three to seven days; 4, lung cancer after the operation after four to six weeks. Box and whisker plots of the measured concentrations of TG-4, TG-7, TG-8, TG-11, TG-19, TG-20, and TG-22 in the exhaled breath 1 h after meals of the study groups, (least significant difference test) LSD-t analysis of variance. * *p* < 0.05; ** *p* < 0.01. (**H**), ROC curve analysis was performed on the exhaled air 1 h after the meal in the control group and the lung cancer group before the operation. ROC curves for the TG-4, TG-7, TG-8, TG-11, TG-19, TG-20, and TG-22 breath model in the diagnosis of lung cancer. The area under the curve of TG-4, TG-7, TG-8, TG-11, TG-19, TG-20, and TG-22 were 0.766, 0.611, 0.636, 0.682, 0.604, 0.611, and 0.625, respectively.

**Figure 2 cancers-15-01186-f002:**
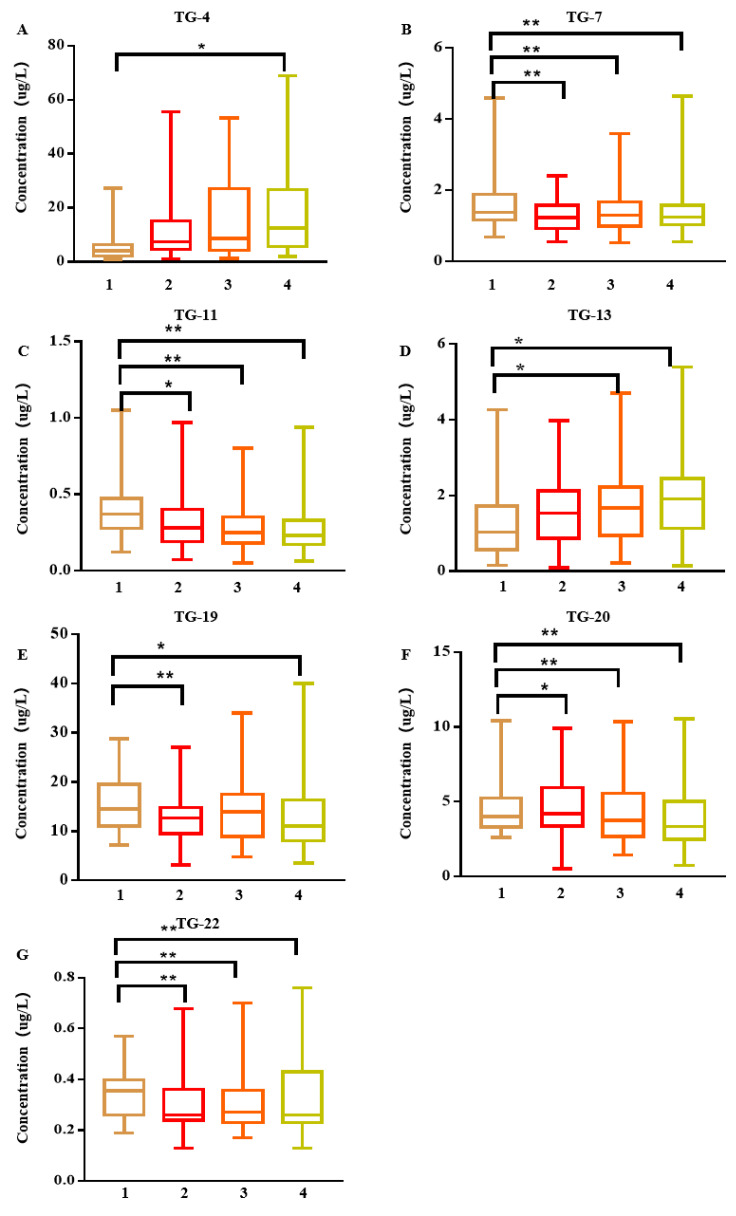
(**A**–**G**), 1, control; 2AIS; 3, MIA; 4, IAC. Box and whisker plots of the measured concentrations of TG-4, TG-7, TG-11, TG-13, TG-19, TG-20, and TG-22 in the exhaled breath 1 h after meals of the study groups, (least significant difference test) LSD-t analysis of variance. * *p* < 0.05; ** *p* < 0.01. (NB: All the data from AIS, MIA, and IAC are before surgery).

**Figure 3 cancers-15-01186-f003:**
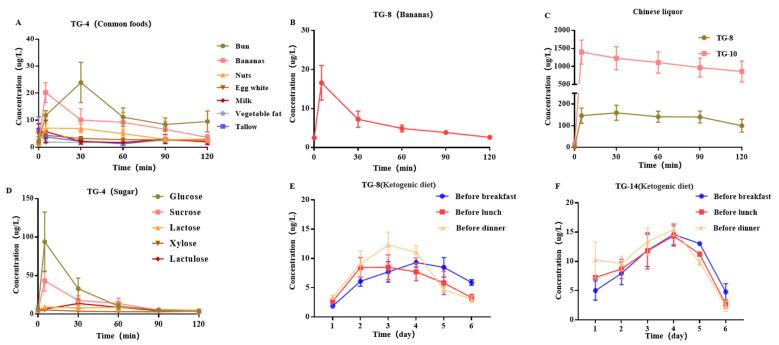
Changes in VOC concentration in exhalation after different diets. (**A**–**C**). Changes in carbonyl VOCs’ concentration in human exhaled breath after eating some common foods. (**D**) Changes in the concentration of TG-4 in the exhaled breath after drinking different sugar solutions (**E**,**F**). Values and error bars represent mean ± SE.

**Figure 4 cancers-15-01186-f004:**
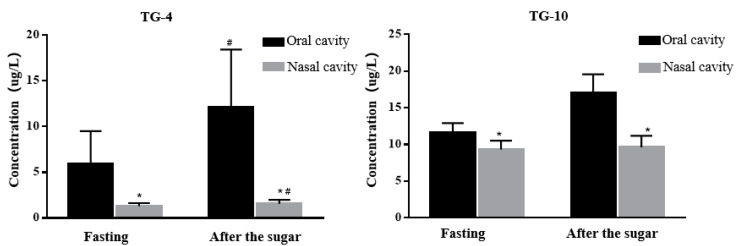
Difference between oral and nasal exhalation. Eight healthy volunteers drank the 5% glucose solution and immediately blew into two sampling bags in parallel with each other in the mouth and nose. Values and error bars represent mean ± SE. * nasal cavity compared with the oral cavity, *p* < 0.05, and # compared fasting after sugar.

**Figure 5 cancers-15-01186-f005:**
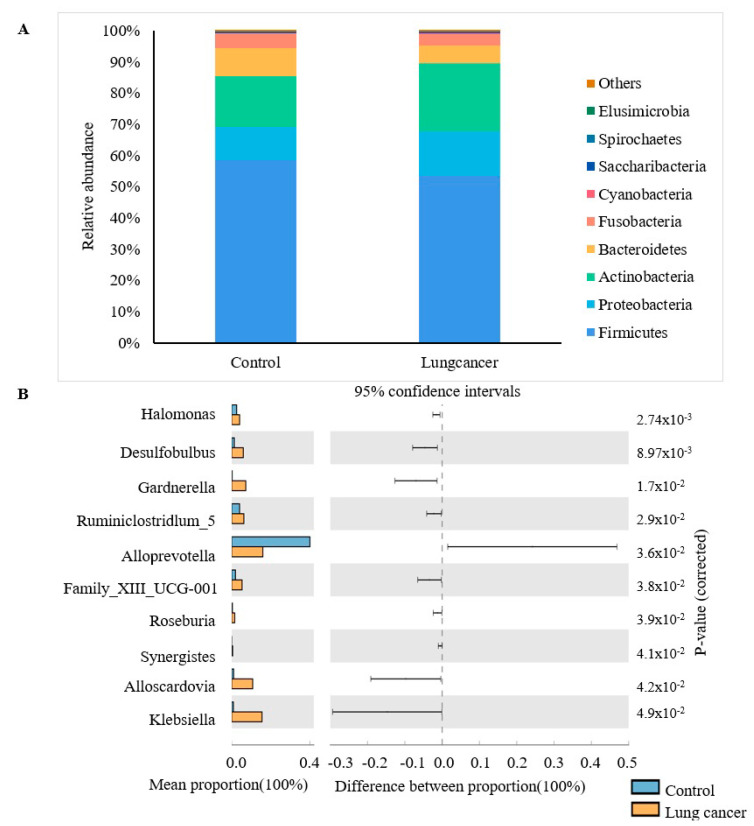
The differences in oral salivary flora between the control group and the lung cancer group. (**A**) At the phylum level, the human oral salivary flora is mainly *Firmicutes, Proteobacteria, Actinobacteria, Bacteroidetes, and Fusobacteria*, accounting for about 99% of the proportion. Compared with the control group, the proportion of Elusimicrobia in the lung cancer group decreased (*p* = 0.038). (**B**) At the genus level, STAMP analyzed the difference in oral saliva between the control group and the lung cancer group.

**Table 1 cancers-15-01186-t001:** Characteristics of the study population.

	Lung Cancer Patients (n = 291)	Healthy Controls (n = 95)	Χ^2^/t	*p* Value
Mean age (years)	60 (19–89)	50 (23–81)	6.917	0.000
Proportion (man)	116 (40%)	27 (29%)	4.02	0.044
Smoking behavior			0.793	0.370
Active	19 (7%)	6 (7%)		
Former	22 (7%)	4 (4%)		
Never	250 (86%)	85 (89%)		
Histologic classification				
Adenocarcinoma	251 (86%)			
Squamous carcinoma	11 (4%)			
Others	29 (10%)			

X^2^/t: Chi-square value divided by *t*-test value.

**Table 2 cancers-15-01186-t002:** Correlation between TG-4 and oral saliva bacterial abundance.

Positive Correlation	Negative Correlation
Phylum: *Spirochetes, Intertrophic bacteria, Pericarpium, Verruca microflora*	Phylum: *Proteobacteria*
Class: *Some spirochetes, Mutual nutrient bacteria, Flexible membrane bacteria, Verruca microbe*	Class: *Saccharibacteria*
Order: *Spirochetes, Ynergistales, Mycoplasmas, Microflora*	Order: *Saccharibacteria*
Family: *Streptococcus, Spirochetes*	Family: *Saccharibacteria*
Genus: *Streptococcus, Treponema Pallidum*	Genus: *Digestive streptococcus, Actinomycetes, Saccharibacteria*
Species: *Streptococcus pharyngitis, Porphyrin monomonas dental medulla, Lactobacillus mucous, Alloscardovia_omnicolens*	Species: *TM7 animal phylum oral clone DR034*

**Table 3 cancers-15-01186-t003:** Previous studies on VOC biomarkers in patients with lung cancer.

Study	Method	VOCs	Sensitivity	Specificity	References
Anton et al.	HS-PTV-MS	2-butanone	40%	100%	[23]
Hanai et al.	GC-TOFMS	2-Pentanone	85%	70%	[24]
G. Song et al.	GC-MS	3-Hydroxy-2-butanone	93%	92.7%	[19]
J. Rudnicka et al.	GC-MS	Acetone	74%	73%	[20]
M. Phillips et al.	GC-MS	Pentane	89.6%	82.9%	[21]
N. Peled et al.	GNP sensor	Toluene	70%	100%	[22]
Mingxiao Li et al.	FT-ICR-MS	2-butanone	97%	84%	[32]
Ralph J. Knipp	FT-ICR-MS and GC-MS	Hydroxyl acetaldehyde and 3-hydroxy-2-butanone	94.2 ± 2.5%	-	[33]
This study	LC-MS/MS	3-Hydroxy-2-butanone	96%	73%	-

GC-MS = gas chromatography mass spectrometry, GC-TOFMS = gas chromatography time of flight mass spectrometry, HS-PTV-MS, headspace-programmed temperature vaporizer-mass spectrometer, GNP = gold nanoparticles, LC-MS/MS = liquid chromatography with tandem mass spectrometry.

## Data Availability

The datasets generated and/or analyzed during the current study are not publicly available due to hospital guidelines and legislation regarding personal data. Data will be available from the corresponding author upon reasonable request and with permission of the Zhoushan Hospital Legal Department.

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
