# Peer review of "LC-MS/MS Based Volatile Organic Compound Biomarkers Analysis for Early Detection of Lung Cancer"

_cancers, 2023, doi:10.3390/cancers15041186_

Round 1

Reviewer 2 Report

I admire the authors for presenting this important issue. I have found this manuscript well-written and structured. However, there are some vital points needed to be added or revised before considering it further. Here are the comments needed to be addressed: 

1. The cellular metabolic processes and their changes are manifested in emitted volatile organic compound (VOC) compositions of different diseased cells, such as cancer cells will open up the possibilities of early detection techniques. The INTRODUCTION section needs information and background on metabolic changes and VOCs and their role in the detection of diseases and cancer. Please also highlight in the section, the other techniques besides LC-MS, as most of them are not feasible to establish in a direct clinical setup. Some recent citations needed to be added, such as the following one from a very nice journal, explaining a technique of insect brain-based cancer VOC detection using AI techniques. 

https://doi.org/10.1016/j.bios.2022.114814 

2. At the end of the introduction please separate the section for the aim and objectives.

Reviewer 3 Report

In the manuscript entitled "LC-MS/MS based volatile organic compound biomarkers anal- 2 ysis for early detection of lung cancer” by Shuaibu Nazifi Sani et al., the authors presented a standardized approach that could be used to screen early lung cancer patients based on VOC markers, overcoming the sensitivity and reproducibility challenges associated with conventional testing methods. The manuscript is very interesting but there are some issues that need to be clarified before acceptance for publication.

Major concern:

The mentioned method has been utilized in different articles. So, the author should mention and clarify the difference between them and the others. Kindly see the following articles:

https://www.sciencedirect.com/science/article/abs/pii/S0169500215300118

https://pubs.rsc.org/en/content/articlelanding/2015/ay/c5ay01576f/unauth

the results are clearly discussed in the manuscript. However, during the discussion, the previously published paper should be compared with the obtained results.

Minor concerns:

Line 20: abbreviation meaning VOC should be mentioned for the first time in the simple summary.

Line 29: abbreviation meaning VOC should be mentioned for the first time in the abstract.

It is better to summarize all abbreviations in a table to clear for the readers.

Line 520: abbreviation of ROC was mentioned previously in the manuscript.

Please revised all abbreviations in the manuscripts for the first time of appearance.

The reference list needs to be updated with the last three years, particularly before 2010.

Round 2
